# Navigating Weight Prediction with Diet Diary

## ABSTRACT

Current research in food analysis primarily concentrates on tasks such as food recognition, recipe retrieval and nutrition estimation from a single image. Nevertheless, there is a significant gap in exploring the impact of food intake on physiological indicators (e.g., weight) over time. This paper addresses this gap by introducing the DietDiary dataset, which encompasses daily dietary diaries and corresponding weight measurements of real users. Furthermore, we propose a novel task of weight prediction with a dietary diary that aims to leverage historical food intake and weight to predict future weights. To tackle this task, we propose a model-agnostic time series forecasting framework. Specifically, we introduce a Unified Meal Representation Learning (UMRL) module to extract representations for each meal. Additionally, we design a diet-aware loss function to associate food intake with weight variations. By conducting experiments on the DietDiary dataset with two state-of-the-art time series forecasting models, NLinear and iTransformer, we demonstrate that our proposed framework achieves superior performance compared to the original models. We will make our dataset, code, and models publicly available.

## CCS CONCEPTS

• **Information systems** → **Multimedia information systems**.

## KEYWORDS

Weight prediction, food analysis, time series forecasting models

## 1 INTRODUCTION

Food plays a vital role in human existence, affecting the quality of life and various physiological indicators, with weight being one of the most fundamental and critical aspects. [24] indicates that the relative balance between energy expenditure and dietary intake determines weight gain or loss. Notably, the energy expenditure in non-exercise activities (such as sitting) usually accounts for a significantly larger portion of total energy expenditure than exercise [21]. Motivated by these insights, we aim to investigate the impact of food intake on weight prediction.

Numerous studies have focused on various tasks within the food domain, such as food classification [11, 25, 35, 37], ingredients recognition [1, 5, 6], recipe retrieval [19, 46, 48, 53, 54, 63, 64], food volume prediction [32, 34, 49] as well as calorie and nutritional information estimation [36, 50, 52]. Nevertheless, these existing methods typically analyze each food image independently and do

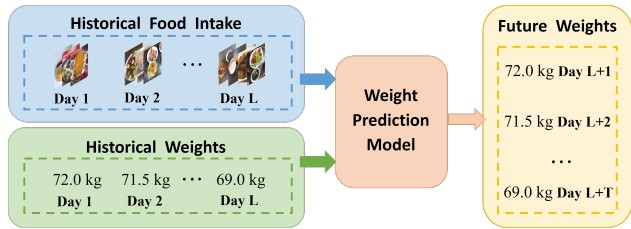

**Figure 1: The overview of the proposed weight prediction with diet diary task. Given a historical food intake and corresponding weight measurement in the past $L$ days, the task aims to predict the weights in the following $T$ days.**

not investigate the impact of food intake on physiological indicators (e.g., weight) over a period of time.

To address this gap, this paper navigates weight prediction with dietary diary. On the one hand, we construct a novel dataset named DietDiary, which includes daily dietary records and corresponding weight measurements for 611 participants from a health management system. Different from existing datasets in the food domain such as [2, 5, 38], DietDiary is the first dataset to provide both food intake and corresponding weight measurements over a period of time. On the other hand, as illustrated in Figure 1, we define a new task that aims to leverage historical weight and food intake data to predict future weights. This task poses two major challenges: (1) Meal representation learning. Given that food intake can be represented in various forms, such as textual ingredient labels or food images, developing an effective method for learning meal representations for weight prediction is crucial. (2) Modeling the correlation between food intake and weight changes. Since food intake is closely related to weight, weight prediction requires more than just exploring the temporal relationships between historical and future weights. Understanding the complex correlation and dependencies between dietary intake and weight fluctuations presents significant challenges to this task.

To address the aforementioned challenges, we propose a novel framework that integrates food intake information for weight prediction. Specifically, we introduce a Unified Meal Representation Learning (UMRL) module that leverages CLIP [43] text or image encoders to extract a unified feature representation of a meal from various forms of historical food intake. Additionally, we propose a diet-aware loss function to enable the model to capture the correlation and dependencies between food and weight changes. Importantly, our framework is designed to be compatible with any existing time series forecasting model for weight prediction that incorporates food intake information. We evaluate our framework on the DietDiary dataset using two representative advanced time series forecasting models, NLinear [59] and iTransformer [31]. The superior performance of our framework over these models demonstrates its effectiveness in leveraging food intake information for weight prediction.

The contributions of this paper can be summarized as follows:

- We construct a new DietDiary dataset, which is the first dataset providing food intake and corresponding weight measurements over a period of time.
- We introduce a novel task of weight prediction using dietary information. This task uniquely treats food intake as temporal data and investigates its potential to improve weight prediction.
- We propose a model-agnostic time series forecasting framework for weight prediction with food intake. Our framework demonstrates significant improvements over existing methods in weight prediction performance.

## 2 RELATED WORK

### 2.1 Time Series Forecasting

Time series forecasting is a fundamental task in various fields, including finance, traffic and meteorology. Over the years, to address the inherent challenges of predicting future values based on past observations, numerous models are proposed, such as the famous ARIMA[3]. With the rise of deep learning, neural network-based approaches have gained popularity in time series forecasting. Models such as Long Short-Term Memory (LSTM) networks[22], DeepAR[44] and Prophet[51] are used to capture long-term dependencies and nonlinear relationships in the data. As Transformer[31] demonstrates powerful sequence modeling capabilities in natural language processing[14], computer vision[4, 15] and other domains, models based on modifying the vanilla Transformer, especially the attention mechanism, tailored for time series forecasting tasks have been widely researched, such as Informer[60], Autoformer [56], Pyraformer [30], and FEDformer [61].

While various research efforts are ongoing to modify attention structures to achieve better Transformer-based solutions, [59] questions the effectiveness of the Transformers and proposes a set of simple and efficient linear models including Linear, NLinear, and DLinear, which have led more researchers to focus on linear-based model. TiDE [13] designs a Multi-layer Perceptron (MLP) based encoder-decoder model to capture covariates and non-linear dependencies. TSMixer [16] patches the time series data and enhances the learning capability of simple MLP structures. Chen et al. [10] propose a Time-mixing MLP to model temporal dependency and a Feature-mixing MLP to analyze the cross-variate information. Influenced by the outstanding performance of linear models, several works continue to employ transformer-based architectures but seek performance enhancement from alternative perspectives. PatchTST [41] uses patched data as input and adopts a channel-independence design that makes a token only contains information from one channel, instead of changing the structure of the transformer. iTransformer [31] inverts the duties of the attention mechanism and the feed-forward layer to capture temporal information to achieve promoted performance and generalization.

In contrast, we propose a model-agnostic time series forecasting framework to predict weight with food intake. To validate the effectiveness of our framework, state-of-the-art linear-based model NLinear [59] and transformer-based model iTransformer [31] are chosen as our time series forecasting models respectively.

### 2.2 Food Analysis

With the development of computer vision and the emergence of various food datasets, research methods and tasks in the food domain have gradually become more diverse. Some traditional visual tasks have been extended to the food domain, such as food classification [11, 25, 27, 35, 37], ingredients recognition [1, 5, 6, 29], and food segmentation [17, 23, 28, 57]. With the release of Recipe1M [47], cross-modal food-related tasks have been extensively studied, such as the retrieval of finding the most relevant recipe for a food image or vice versa [8, 9, 19, 48, 63]. Some works focus on generating recipes from images [12, 20, 45] or corresponding images from recipes [42, 62]. These tasks all involve analyzing each image independently, whereas in this paper, we model temporal dietary data, which has not been explored in previous studies.

Another research branch involves predicting the nutritional components and calorie of food from food images, which aids in monitoring intake patterns. Some works estimate volume first from voxel [36], point cloud [18, 33] or 3D mesh [40], then mapping is achieved through data on the calories contained per unit volume. With the release of Nutrition5K [52], a dataset providing fine-grained nutritional attributes, food quality and food videos, some works predict nutrition directly from images by neural network [50, 52, 58]. However, these works only predict nutritional information without further linking food intake to weight. To the best of our knowledge, this paper is the first work to utilize dietary diary to predict weight.

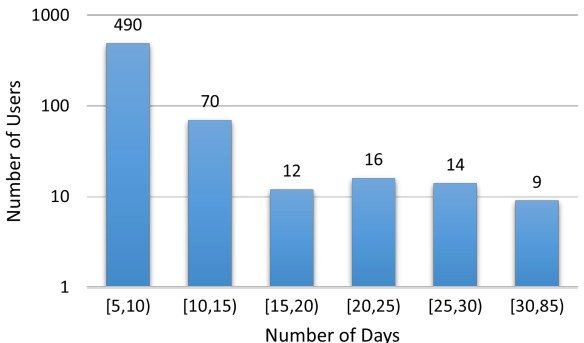

**Figure 2: Distribution of the number of recording days for the participants in DietDiary dataset. The y-axis is in the log scale.**

## 3 DATASET CONSTRUCTION

We introduce a novel dataset, DietDiary, specifically for analyzing weight in relation to food intake. In contrast to datasets such as Food-101 [2], VIREO Food-172 [5, 7], Food2K [39] and Recipe1M [47], DietDiary encompasses diet diary of three meals over a period of time, accompanied by daily weight measurement. To the best of our knowledge, DietDiary is the first dataset in the food domain to provide this kind of data, offering new opportunities for research in dietary pattern analysis and its impact on weight management.

breakfast          lunch          supper

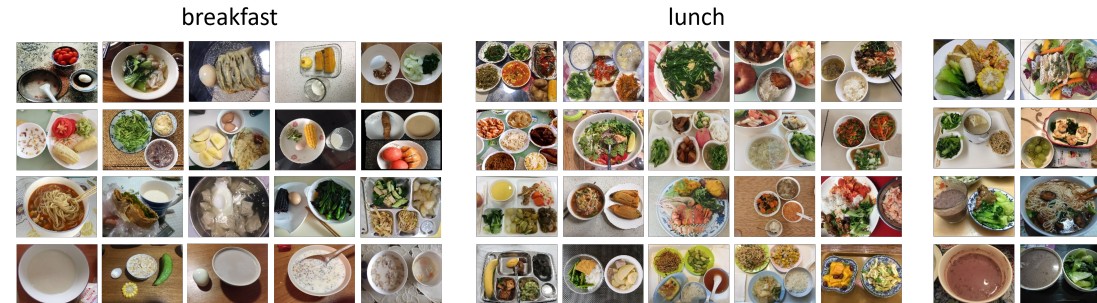

**Figure 3: Examples of food images from three meals in DietDiary.**

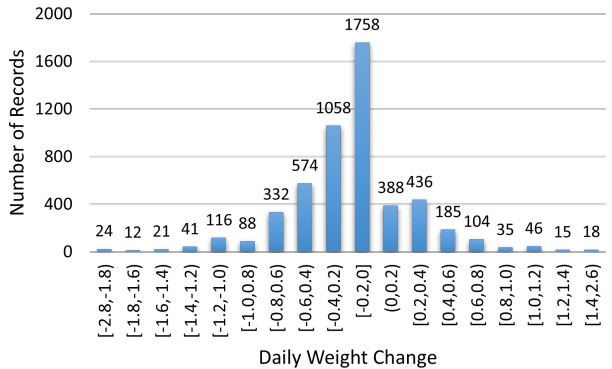

**Figure 4: Distribution of Daily Weight Change of all records in DietDiary.**

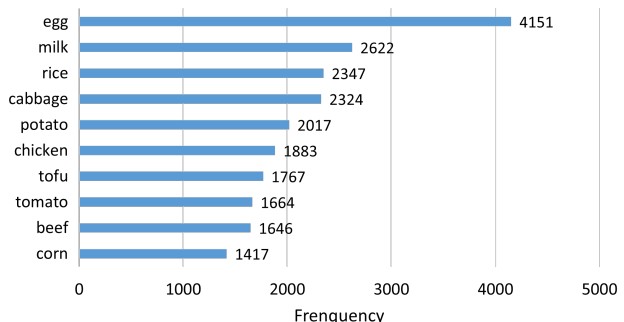

**Figure 5: Top 10 ingredients by occurrence frequency in DietDiary.**

### 3.1 DietDiary Dataset

The DietDiary dataset was collected within a health management system [1] wherein participants were required to meticulously log their daily dietary intake, along with their corresponding weight measurements. The dietary log consists of images of three meals (i.e., breakfast, lunch, and supper) along with manually labeled ingredients for each meal. The dataset encompasses a total of 611

_______________
[1]https://www.qiezilife.com/

participants, with over 5k daily records, nearly 30k images, and over 15k ingredient annotations.

As depicted in Figure 2, the duration of record-keeping among participants varies, ranging from a minimum of one week to over one month. A significant majority (91.6%) of the participants maintained their records for a duration of one to two weeks. However, there were also 9 participants recorded their diet and weight for more than a month. Figure 4 illustrates the distribution of daily weight fluctuations across all participants. Most users experience weight changes within ±1 kg per day, with weight loss records comprising the majority due to the data source. Additionally, a small portion of individuals experience changes exceeding 2 kg within a day.

### 3.2 Food Images and Ingredient Annotation

Figure 3 presents meal images from various participants, illustrating typical examples of breakfast, lunch, and supper. It is important to note that we have manually curated the dataset to exclude images that are not relevant to food.

The annotation of ingredients in the dataset was standardized through a series of preprocessing steps. First, we unified the separators used between phrases, as different participants employed varying symbols (e.g., "- ", " / ", " () ") to delineate ingredients. Second, the original annotations in Chinese were translated into English. Third, we adopted the approach as in inverse cooking [45] to aggregate and categorize ingredients based on common prefixes or suffixes, for example, "boiled eggs", "steamed eggs", and "egg slices" are all merged into the "egg" category. Fourth, ingredients that appeared less than five times were excluded from the dataset. After preprocessing, the total number of unique ingredients was 197. The ten most frequently occurring ingredients are depicted in Figure 5, such as "milk", "egg" and "rice", all of which are common in everyday meals. Furthermore, Figure 6 provides a comprehensive example from the dataset, showcasing food images, annotated ingredients, and corresponding weight measurements for two participants with different weight fluctuation trends. On the one hand, user A maintained a healthy dietary habit, and the weight measurements consistently show a decreasing trend. On the other hand, user B consumed high-calorie foods (such as fried shrimp and sugar) on the third and fifth days, and also consumed a large amount of staple food (dumplings and sushi) during supper, resulting in an increase in corresponding weight records.

| Day | User | Breakfast | Lunch | Supper | Weight(kg) | User | Breakfast | Lunch | Supper | Weight(kg) |
|-----|------|-----------|-------|--------|------------|------|-----------|-------|--------|------------|
| 1 | | chicken, bread, lettuce, potato, milk | chicken, bean, rice, carrot, curry, potato | cabbage, tofu, mushroom, lettuce | 101.0 - 100.3 | | grain, porridge, meat, egg | pepper, chicken, cuttlefish, green, rice | cucumber, noodle, bamboo_shoot | 66.7 − 66.5 |
| 2 | | chicken, corn, cucumber kale, lettuce, milk, egg | cauliflower, rice, beef carrot, celery, peanut | cabbage, carrot | 100.3 - 100.1 | | peanut, porridge, cucumber, egg | tofu, shrimp, green, rice | meat, mushroom, noodle | 66.5 - 66.1 |
| 3 | A | broccoli, milk, egg | wood-ear, noodle, mushroom, egg | meat, shrimp, lettuce | 100.1 - 99.5 | B | meat, egg, noodle | meat, rice | fried_shrimp, seafood | 66.1 - 66.5 |
| 4 | | chicken, corn, broccoli, bread, tofu, milk ,egg | bean, pizza, lettuce | tofu, shrimp, soup, cucumber | 99.5 - 99.3 | | milk, egg | salad, green, shellfish potato | shellfish | 66.5 − 65.9 |
| 5 | | tomato, egg | pepper, bean, rice carrot, beef, oatmeal | shrimp, cucumber, tomato, lettuce, soup | 99.3 - 99.1 | | chicken, bean, melon pumpkin, egg | shrimp, sugar, rice lotus_root | salad, dumpling, sushi | 65.9 - 66.0 |
| ... | | ... | ... | ... | ... | | ... | ... | ... | ... |

**Figure 6: Data records for two participants with different weight fluctuation trends in DietDiary. The records leading to weight gain are highlighted in red.**

## 4 METHOD

In this section, we provide detailed definition of the task in Section 4.1 and elaborate on the details of our proposed framework in Sections 4.2 and 4.3.

### 4.1 Task Definition

Given the historical weight sequence $W = \{w_t\}_{t=1}^{L}$ and corresponding food intake $F = \{Breakfast_t, Lunch_t, Supper_t\}_{t=1}^{L}$ for the past $L$ days, the goal of our proposed task is to predict the future weight sequence $\hat{W} = \{w_t\}_{t=L+1}^{L+T}$ over the next $T$ days, where $w_t$ denotes the weight value of the t-*th* day, $Breakfast_t$, $Lunch_t$ and $Supper_t$ represent sets of food images or ingredient labels for each meal on the t-*th* day. Note that the historical observation takes various data modalities, including textual ingredient labels or food images, and numerical weight.

### 4.2 Proposed Framework

Figure 7 depicts an overview of the proposed framework. The framework primarily consists of three key components: a unified Meal Representation Learning (UMRL) module, an agnostic time series forecasting model, and a diet-aware loss computation.

**Unified Meal Representation Learning (UMRL)** is a module to extract representation for each meal. Given the historical food intake (e.g., food images or ingredient labels), UMRL is designed to obtain representation for breakfast, lunch, and supper for each day respectively, denoted as $f = \{b_t, l_t, s_t\}_{t=1}^{L}$, where $b_t$, $l_t$ and $s_t$ are representation for breakfast, lunch, and supper on the t-*th* day. More details of UMRL are presented in Section 4.3.

**Model-agnostic weight prediction.** After obtaining the representation of historical food intake from UMRL, we concatenate it with the historical weight $W$, denoted as $X = \{b_t, l_t, s_t, w_t\}_{t=1}^{L}$, and feed into a time series forecasting model $M$. The weight prediction result is obtained as follows:

$$\hat{Y} = \{\hat{b}_t, \hat{l}_t, \hat{s}_t, \hat{w}_t\}_{t=L+1}^{L+T} = M(X), \tag{1}$$

where $\hat{b}_t, \hat{l}_t, \hat{s}_t, \hat{w}_t$ are the predicted weights from breakfast, lunch, supper and weight respectively. Note that our proposed framework is agnostic to the time series forecasting model, which gives us the flexibility to utilize existing state-of-the-art models for our task. In this paper, we choose linear-based NLinear [59] and Transformer-based iTransformer [31] as $M$ in Equation 1.

**Diet-aware loss computation.** We introduce a diet-aware loss to model the food intake and weight variations. Assume the weight variation $\Delta_t$ of the t-*th* day is calculated by the following formula:

$$\Delta_t = w_t - w_{t-1}. \tag{2}$$

As each meal contributes to the weight variations, the diet loss is computed by combining all three meals as follows:

$$\mathcal{L}_{diet} = \frac{1}{3}((\Delta_t - \hat{b}_t)^2 + (\Delta_t - \hat{l}_t)^2 + (\Delta_t - \hat{s}_t)^2). \tag{3}$$

In addition, we compute a weight loss based on the weight prediction from historical weight as follows:

$$\mathcal{L}_{weight} = (w_t - \hat{w}_t)^2. \tag{4}$$

The overall diet-aware loss to train the proposed framework is to combine both diet and weight losses:

$$\mathcal{L} = \lambda \mathcal{L}_{weight} + (1 - \lambda) \mathcal{L}_{diet}, \tag{5}$$

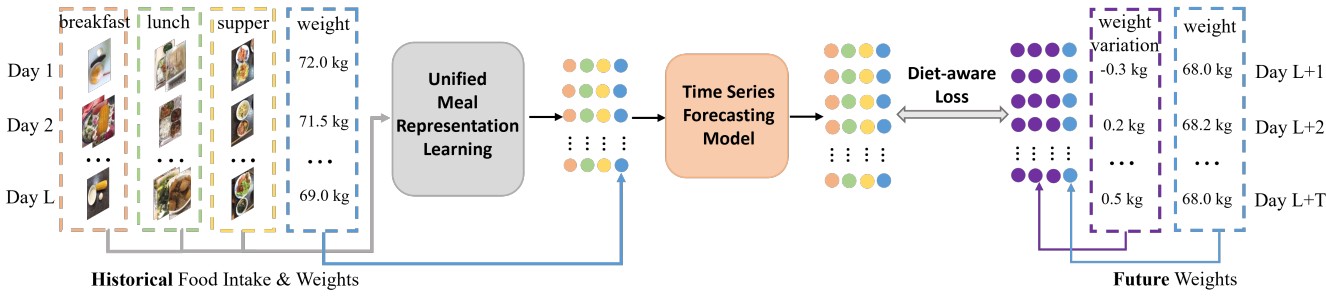

**Figure 7: Framework Overview. The "Unified Meal Representation Learning" module is proposed to map the historical food intake into a time series meal feature sequence. The features and historical weight sequence are combined and subsequently fed into an agnostic time series forecasting model to predict future weights.**

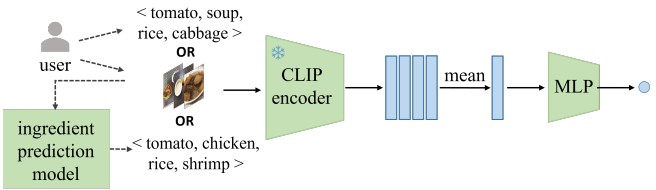

**Figure 8: The architecture of proposed Unified Meal Representation Learning Module.**

where $\lambda$ is a hyper-parameter to balance the two losses.

## 4.3 Unified Meal Representation Learning

As shown in Figure 8, we propose Unified Meal Representation Learning (UMRL) Module to extract representation from historical food intake for each meal. As the food intake could be either food images or ingredient labels, we first employ CLIP [43] to encode both visual or textual inputs to unify the feature extraction. On the one hand, if the historical food intake is ingredient label, we adopt pretrained CLIP text encoder to extract the features. On the other hand, we adopt CLIP image encoder to extract the features from historical food images. The CLIP model is kept frozen during our training. Note that the users prefer taking pictures of their meals rather than typing the concrete ingredient annotations in practice. As a result, we also investigate employing a pre-trained ingredient recognition model to automatically obtain ingredient labels from images, which is illustrated in the bottom left of Figure 8.

The CLIP features of images or ingredient labels are subsequently averaged to get a feature vector for each meal. Finally, we feed this vector into a Multilayer Perceptron (MLP) to derive the final meal representation. Take breakfast food images as an example, denoted $Breakfast_t = \{img_1, ..., img_n\}_{n=1}^{N}$ as a set of breakfast food images on the $t$-*th* day, where $N$ is the number of images, the process of UMRL can be formalized as follows:

**Table 1: Statistics of training, testing, and validation sets for different settings. L-T refers to the setting of using weight and food intake in $L$ history days to predict the weights of future $T$ days.**

| Data split | settings (L-T) | | | | |
|---|---|---|---|---|---|
| | 3-3 | 3-5 | 3-7/5-5/7-3 | 5-7 | 7-7 |
| train | 1,535 | 1,010 | 837 | 672 | 514 |
| validation | 221 | 149 | 123 | 74 | 57 |
| test | 440 | 296 | 241 | 216 | 164 |

$$E_n = CLIPEncoder(img_n),$$

$$\bar{E} = \frac{1}{N}\sum_{n=1}^{N}(E_n), \quad (6)$$

$$b_t = MLP(\bar{E}),$$

where $E_n$ is the CLIP image feature for $img_n$, $\bar{E}$ indicates the averaged CLIP image features and $b_t$ is the breakfast representation.

## 5 EXPERIMENT

### 5.1 Experiment Settings

**Dataset.** We conduct experiments on our DietDiary dataset. Inspired by time series forcasting task [31, 59], we extensively explore different combinations of using weight and food intake history in $L$ days to predict the weights in future $T$ days, denoted as setting L-T. The settings examined include {3-3, 3-5, 3-7, 5-5, 5-7, 7-3 and 7-7}. For instance, in the 7-3 setting ($L = 7$, $T = 3$), we utilize the historical weight sequence and corresponding food intake from the past 7 days to predict the weight for the subsequent 3 days during training. The dataset is partitioned into training, validation, and test sets in a 7:1:2 ratio for each setting. It is crucial to ensure that each participant appears exclusively in one of these splits, thereby preventing any overlap of participants across training, validation, and testing sets. The settings determine the minimum number of days required for training. To maximize the utility of the dataset, we employ different data partitions for different settings. For example, the settings 5-5 and 7-3 necessitate a minimum of 10 days of dietary intake records per participant, whereas the 7-7 setting

**Table 2: Performance comparison based on NLinear as time series forecasting model in terms of MSE and MAE. "image", "ing-users" and "ing-LMM" represent the food intake is food image, ingredient labels provided by users, ingredient labels predicted by FoodLMM respectively.**

| setting | 3 - 3 | | 3 - 5 | | 3 - 7 | | 5 - 5 | | 5 - 7 | | 7 - 3 | | 7 - 7 | |
|---|---|---|---|---|---|---|---|---|---|---|---|---|---|---|
| metric | MSE | MAE | MSE | MAE | MSE | MAE | MSE | MAE | MSE | MAE | MSE | MAE | MSE | MAE |
| NLinear [59] | 3.729 | 1.107 | 5.390 | 1.448 | 6.949 | 1.710 | 5.991 | 1.619 | 5.331 | 1.589 | 5.464 | 1.600 | 6.791 | 1.958 |
| ours image | 2.769 | 0.985 | **4.765** | **1.371** | 6.332 | 1.636 | 2.573 | **1.330** | 3.219 | **1.341** | 2.799 | 1.213 | **2.454** | 1.376 |
| ours ing-users | 2.865 | 0.998 | 4.819 | 1.377 | **6.307** | **1.633** | 2.539 | 1.333 | 3.235 | **1.341** | 2.742 | **1.211** | 2.479 | 1.370 |
| ours ing-LMM | **2.048** | **0.962** | 4.849 | 1.380 | 6.312 | 1.635 | 2.542 | 1.345 | 3.247 | 1.342 | 2.968 | 1.233 | 2.497 | **1.365** |

**Table 3: Performance comparison based on iTransformer as time series forecasting model. "image", "ing-users" and "ing-LMM" represent the food intake is food image, ingredient labels provided by users, ingredient labels predicted by FoodLMM respectively.**

| setting | 3 - 3 | | 3 - 5 | | 3 - 7 | | 5 - 5 | | 5 - 7 | | 7 - 3 | | 7 - 7 | |
|---|---|---|---|---|---|---|---|---|---|---|---|---|---|---|
| metric | MSE | MAE | MSE | MAE | MSE | MAE | MSE | MAE | MSE | MAE | MSE | MAE | MSE | MAE |
| iTransformer [31] | 4.023 | 1.402 | 5.306 | 1.717 | 5.918 | 1.866 | 5.711 | 1.817 | 6.369 | 1.966 | 4.657 | 1.611 | 4.851 | 1.701 |
| ours image | **3.436** | **1.268** | 5.268 | 1.710 | 5.791 | 1.835 | 5.478 | 1.783 | 4.299 | 1.616 | 4.524 | 1.584 | **4.411** | **1.596** |
| ours ing-users | 3.544 | 1.293 | 5.202 | 1.696 | 5.922 | 1.861 | **3.540** | **1.466** | 3.746 | 1.541 | 4.322 | 1.545 | 4.645 | 1.641 |
| ours ing-LMM | 4.133 | 1.424 | **5.047** | **1.662** | **5.637** | **1.802** | 4.626 | 1.611 | **3.657** | **1.532** | **3.992** | **1.486** | 4.452 | 1.610 |

requires at least 14 days. Table 1 presents the dataset statistics for various settings.

**Evaluation metrics.** We employ the Mean Squared Error (MSE) and Mean Absolute Error (MAE) as metrics for performance evaluation. Given that users are primarily concerned with long-term trends in weight prediction, we evaluate performance through autoregressive weight prediction. Specifically, during testing, given a history of $L$ days, we initially forecast the weights for the subsequent $T$ days. Subsequent predictions are then made autoregressively, using the outcomes of previous forecasts as input, until predictions have been generated for all recorded days of users in the test set.

**Implementation details.** Given the remarkable performance of FoodLMM [58], a Large Multi-modal Model tailored in the food domain, we choose FoodLMM as our ingredient prediction model. For convenience, we use "users" to refer to ingredients annotated by users and "LMM" to refer to ingredients predicted by the FoodLMM. We evaluate our framework on both the NLinear [59] model, representing linear time series forcasting models, and the iTransformer [31] model, representing transformer-based solutions. For iTransformer, the dimension of series representation $D$ is set to 16 when $L \in \{3, 5\}$, 32 when $L \in \{5, 7\}$. The dimension of feed-forward layer is twice that of $D$. The number of inverted Transformer blocks and attention heads are set to 1 and 2 respectively. Both NLinear and iTransformer are trained using Adam optimizer [26] with batch size of 32 until early-stopping criteria is met (validation loss does not decrease after 7 epochs). The learning rate is initially set to 0.005 and decays with each epoch. The $\lambda$ in equation 5 is set to {0, 0.1, 0.25, 0.5, 0.75, 1} to explore the influence of the proportion of diet loss, which will be shown in Section 5.4.

## 5.2 Performance Comparison

Tables 2 and 3 present the performance of our proposed method, which leverages state-of-the-art time series forecasting models NLinear [59] and iTransformer [31], respectively. Compared to models that do not incorporate food intake information, our method consistently achieves superior performance over NLinear [59] and iTransformer [31] across all evaluated settings, with a significant margin of improvement. Notably, the improvements are observed consistently with three different types of food intake inputs: food images, user-provided ingredient labels, and ingredient labels predicted by FoodLMM. These results clearly demonstrate the effectiveness of incorporating food intake information in enhancing the accuracy of weight prediction.

An inspiring observation is that the ingredient labels predicted by FoodLMM often perform comparably or even surpass manually provided ingredient labels by users in most settings. This finding is noteworthy considering that acquiring food images is generally easier than manually annotating ingredients for each meal. Figure 9 presents four examples comparing ingredient labels between users and FoodLMM. Typically, users are more prone to omitting ingredients rather than mislabeling them. For instance, "noodles", "Chinese parsleycoriander ", and "green onion" in Figure 9 (a), and "corn" and "Shanghai cabbage" in Figure 9 (b) are missing from the user-provided labels. Remarkably, FoodLMM successfully identifies these omitted ingredients. However, ingredient recognition remains a challenging task [7], and the predictions from FoodLMM are still not perfect. For example, there are three incorrect ingredients predicted by FoodLMM in Figure 9 (b) and (c). In Figure 9 (d), although FoodLMM fails to accurately predict "tomato", the remaining ingredients are correctly identified.

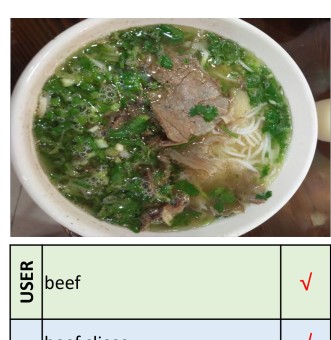 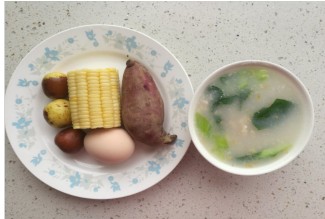 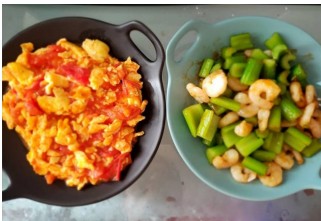

| USER | beef | √ |
|------|------|---|
| | beef slices | √ |
| FoodLMM | noodles | √ + |
| | Chinese parsley/coriander | √ + |
| | minced green onion | √ + |
| | water | √ + |

(a)

| USER | egg | √ |
|------|------|---|
| | date | √ |
| | porridge | √ |
| | potato | √ |
| FoodLMM | boild egg slices | √ |
| | corn blocks | √ + |
| | Shanghai cabbage | √ + |
| | hob blocks of radish | × |
| | lentinus edodes | × |
| | noodles | × |

(b)

| USER | cabbage | √ |
|------|------|---|
| | tofu | √ |
| | chicken | √ |
| | pumpkin | √ |
| FoodLMM | green vegetables | √ |
| | tofu chunks | √ |
| | minced green onion | × |
| | mutton slices | × |
| | sliced carrot | × |

(c)

| USER | shrimp | √ |
|------|------|---|
| | celery | √ |
| | egg | √ |
| | tomato | √ |
| FoodLMM | shelled fresh shrimps | √ |
| | celery | √ |
| | scrambled egg | √ |

(d)

**Figure 9: Examples of ingredient labels from user (green) and LMM (blue) for the same food image. The "√" sign means the correct ingredients, while the "×" sign means the incorrect. The "√+" sign indicates the ingredients from LMM supplement those missed by users.**

**Table 4: The multi-modal fusion results of images and ingredient annotations from users based on NLinear.**

| setting | | 5 - 5 | | 5 - 7 | | 7 - 3 | | 7 - 7 | |
|---------|---------|-------|-------|-------|-------|-------|-------|-------|-------|
| metric | | MSE | MAE | MSE | MAE | MSE | MAE | MSE | MAE |
| NLinear [59] | | 5.991 | 1.619 | 5.331 | 1.589 | 5.464 | 1.600 | 6.791 | 1.958 |
| ours | image | 2.573 | 1.330 | 2.751 | 1.469 | 3.722 | 1.340 | 4.390 | 1.576 |
| | ing-users | **2.539** | 1.333 | 2.773 | 1.480 | 4.327 | 1.430 | 4.340 | 1.567 |
| | fusion | 2.567 | **1.319** | **2.748** | **1.464** | **3.623** | **1.323** | **4.334** | **1.566** |

**Table 5: The multi-modal fusion results of images and ingredient annotations from FoodLMM based on NLinear.**

| setting | | 5 - 5 | | 5 - 7 | | 7 - 3 | | 7 - 7 | |
|---------|---------|-------|-------|-------|-------|-------|-------|-------|-------|
| metric | | MSE | MAE | MSE | MAE | MSE | MAE | MSE | MAE |
| NLinear [59] | | 5.991 | 1.619 | 5.331 | 1.589 | 5.464 | 1.600 | 6.791 | 1.958 |
| ours | image | 2.573 | 1.330 | 2.751 | 1.469 | 3.722 | 1.340 | 4.390 | 1.576 |
| | ing-LMM | **2.542** | 1.345 | 2.765 | 1.476 | 4.523 | 1.461 | 4.367 | 1.572 |
| | fusion | 2.563 | **1.325** | **2.741** | **1.460** | **3.653** | **1.328** | **4.361** | **1.571** |

## 5.3 Multi-modal Fusion of Food Intake

We further investigate multi-modal fusions of food intake from both images and ingredient labels by averaging the UMRs of the same meal across different modalities. Tables 4 and 5 present the fusion results of images with ingredient annotations from users and FoodLMM respectively. Except for the MSE of setting 5-5, the fusion results consistently outperform those of single modality, whether the ingredient labels are manually provided or predicted by FoodLMM. This demonstrates that food information from different modalities can complement each other, which leads to incorporating both of them achieving better performance. Additionally, the effectiveness of multi-modal fusion also validates that our proposed UMRL module can embed food images and ingredient annotations into a common space and learn unified representations of meals from different modalities.

## 5.4 Ablation study

**Impact of number of meals.** Table 6 presents the impact of incorporating different numbers of daily meals on weight prediction performance. Generally, the inclusion of food intake information, regardless of the number of meals, manages to improve performance. When only one meal is considered, the inclusion of lunch

**Table 6: Ablation study of number of meals as food intake based on NLinear as time series forecasting model. Food images are used as food intake and MAE is reported.**

| setting | 3-3 | 3-5 | 3-7 | 5-7 |
|---------|-----|-----|-----|-----|
| NLinear [59] | 1.107 | 1.448 | 1.710 | 1.589 |
| + breakfast | 0.990 | 1.407 | 1.701 | 1.452 |
| + lunch | 0.986 | 1.408 | 1.692 | 1.450 |
| + supper | 1.032 | 1.413 | 1.697 | 1.450 |
| + breakfast + lunch | 1.004 | 1.404 | 1.695 | 1.375 |
| + breakfast + supper | 0.987 | 1.424 | 1.702 | 1.373 |
| + lunch + supper | 1.012 | 1.410 | 1.686 | 1.374 |
| + all three meals | **0.985** | **1.371** | **1.636** | **1.341** |

data yields the most significant improvement under most settings. This is attributed to lunch meals providing the most diverse and distinct food information within the dataset. Breakfast and supper, often featuring less distinctive items like congee-like foods (e.g., meal replacement powders), do not contribute as significantly as lunch to the encoder's effectiveness. For example, the images in the last row of both breakfast and supper in Figure 3. However,

the combination of all three meals offers complementary benefits, leading to the best overall results for weight prediction.

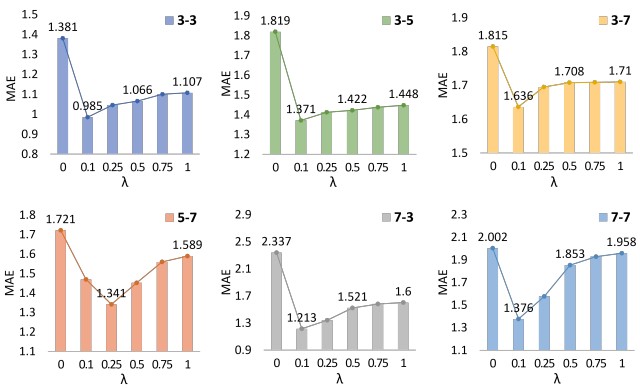

**Figure 10: Ablation study of hyper-parameter of $\lambda$ based on NLinear model in terms of MAE.**

**Hyper-parameter $\lambda$.** We further conduct an ablation study to investigate the impact of the hyper-parameter $\lambda$ in Equation 5, which balances the contributions of the weight loss $\mathcal{L}_{weight}$ and the diet loss $\mathcal{L}_{diet}$. Figure 10 illustrates the performance of weight prediction under different settings with $\lambda$ values ranging from 0 to 1, specifically $\lambda \in \{0, 0.1, 0.25, 0.5, 0.75, 1\}$. As $\lambda$ increases from 0 to 1, which corresponds to a decreasing emphasis on $\mathcal{L}_{diet}$, we observe an initial improvement in performance followed by a gradual decline. The best results are achieved at $\lambda = 0.1$ in most of the settings, demonstrating the robustness of the hyper-parameters across settings. Notably, when $\lambda = 1$, only $\mathcal{L}_{weight}$ is used to optimize the model, which is equivalent to NLinear model by excluding food intake. On the contrary, when $\lambda = 0$ and the model is solely optimized using $\mathcal{L}_{diet}$, resulting in inferior performance by combining the weight loss. The results show the significance of integrating both weight and diet losses for effective weight prediction.

### 5.5 Weight Prediction Visualization

As illustrated in Figure 11, we qualitatively compare the weight prediction visualization among ground-truth weight (blue), NLinear model (green), and our framework based on NLinear (orange) using images as dietary information for different users. It is evident that both the trend and the exact predicted values of our framework are closer to the ground truth than those of the NLinear model. For instance, in subplot (b), the ground truth weight trend shows an initial decrease, followed by a slight increase, and then another decrease. The orange line, representing our prediction results, exhibits the same trend and is numerically closer to the ground truth than the green line of the NLinear model. Similar trends can be observed in subplots (a), (d), (f), and (g) as well. However, in subplots (c) and (i), even though our prediction results are superior to the NLinear model, our framework's predictions do not align well with the ground truth trend. This discrepancy can be attributed to variable changes in weight and the accumulation of prediction errors in long-term predictions. This issue remains an unresolved challenge in time series forecasting tasks [55, 59].

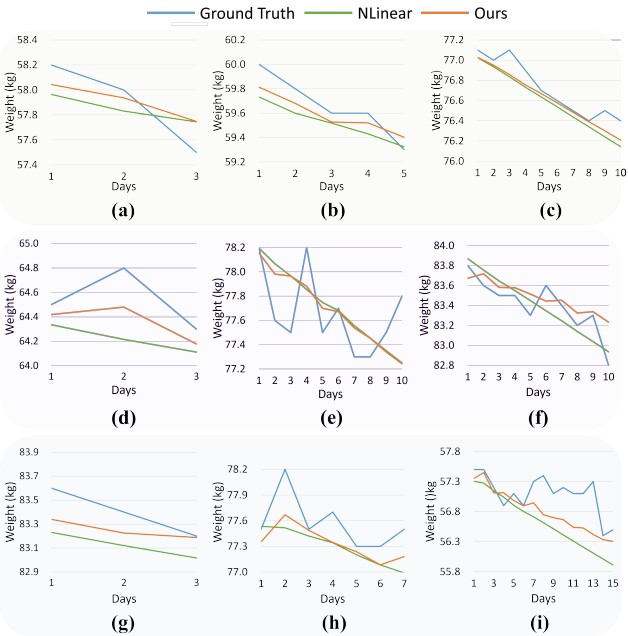

**Figure 11: Weight prediction visualization. In each subplot, the x-axis represents the number of predicted days, and the y-axis refers to the weight(kg). The settings for 3-3, 5-5, and 7-3 are displayed in the first (i.e., (a), (b) and (c)), second (i.e., (d), (e) and (f)), and third (i.e., (g), (h) and (i)) rows respectively.**

## 6 CONCLUSION

We have investigated weight prediction using diet diary. A new dataset named DietDiary is constructed which comprises dietary intake and corresponding daily weight measurements over a period. We introduce a new task of predicting weights by leveraging historical food intake. To address this task, we propose a model-agnostic time series forecasting framework that achieves significant improvements in weight prediction. Our experiments highlight the effectiveness of the proposed UMRL module and diet-aware loss in learning unified intake representation and establishing a correlation between food intake and weight change. Furthermore, we examine three forms of food intake for weight prediction, including images, ingredient annotations provided by users, and ingredient labels predicted from a pre-trained ingredient prediction model. Experimental results on two representative time series prediction models, NLinear [59] and iTransformer [31], demonstrate that incorporating food intake leads to improved accuracy in weight prediction across all three forms. We also investigate multi-modal fusions of dietary information from different modalities (image and text) and achieve better performance, which demonstrates that multi-modal food information can complement each other and provide more effective food information. Through ablation studies, we further demonstrate the necessity of using all three meals for accurate weight prediction. While encouraging, this paper primarily focuses on weight prediction from food intake, exploring the impact of food to other physiological indicator such as blood glucose, will be our future work.

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
