# OpenReview forum: "Navigating Weight Prediction with Diet Diary"
_acmmm.org/ACMMM/2024/Conference — MM2024 Oral_

### Official Review · Reviewer_daWb · 2024-05-23

**Rating:** 5
**Confidence:** 4

**Summary:**

This paper presents a simple but novel approach to weight prediction using dietary diaries. The authors introduce the DietDiary dataset, which contains daily dietary records and corresponding weight measurements from real users. They propose a model-agnostic time series forecasting framework incorporating a Unified Meal Representation Learning (UMRL) module and a diet-aware loss function to leverage historical food intake for predicting future weights. The framework is evaluated using two advanced time series forecasting models, NLinear and iTransformer, demonstrating superior performance over baseline models.

**Strengths:**

1. The introduction of the DietDiary dataset and the task of weight prediction using dietary information are both interesting. This fills a gap in food analysis research, which has traditionally focused on tasks like food recognition and nutrition estimation from images.
2. The proposed UMRL module and diet-aware loss function are innovative components that effectively integrate food intake information into the weight prediction task. This approach is both model-agnostic and adaptable, making it a versatile solution for various time series forecasting models.
3. The comprehensive experimental setup and the use of state-of-the-art models (NLinear and iTransformer) for validation demonstrate the effectiveness of the proposed framework. The results show significant improvements in weight prediction accuracy when incorporating dietary information.
4. This research has practical implications for health management and dietary monitoring, potentially aiding individuals in better understanding the impact of their dietary habits on their weight over time.

**Limitations:**

1. While the DietDiary dataset is novel, it is limited in size (611 participants) and duration (mostly 1-2 weeks). This may affect the generalizability of the results to larger and more diverse populations over longer periods. In addition, it is recommended to put the specific data size into abstract.
2. The evaluation primarily focuses on the accuracy of weight prediction. However, the paper could benefit from additional analysis, such as the impact of different types of meals (e.g., high-calorie vs. low-calorie) on weight prediction accuracy.
3. The paper is generally well-written and organized, but there are some areas where clarity could be improved, particularly in explaining more technical details of the UMRL module.
4. The literature review is comprehensive, but some key references on recent advancements in food analysis could be added to provide a more complete context. For example, Min, Weiqing, et al. "A survey on food computing." ACM Computing Surveys (CSUR) 52.5 (2019): 1-36; Zhou, Pengfei, et al. "Synthesizing Knowledge-enhanced Features for Real-world Zero-shot Food Detection." IEEE Transactions on Image Processing (2024).
5. The references [37]-[39] are repeated, please double-check the presentation details in this paper.

**Suitability:**

3

---

### Official Review · Reviewer_Tz7x · 2024-05-25

**Rating:** 4
**Confidence:** 3

**Summary:**

The paper introduces the DietDiary dataset, which comprises daily dietary diaries and corresponding weight measurements of real users. It introduces a novel task of weight prediction utilizing a dietary diary, aiming to leverage historical food intake and weight to predict future weights. To address this task, the paper proposes a model-agnostic time series forecasting framework. Specifically, it introduces a Unified Meal Representation Learning (UMRL) module to extract representations for each meal and designs a diet-aware loss function to associate food intake with weight variations. Through experiments on the DietDiary dataset using two state-of-the-art time series forecasting models, NLinear and iTransformer, the paper demonstrates that the proposed framework outperforms the original models.

**Strengths:**

1. Novelty and Theoretical Approach: The paper onstruct a new DietDiary dataset, which is the first dataset providing food intake and corresponding weight measurements over a period of time. The paper introduce a novel task of weight prediction using dietary information. This task uniquely treats food intake as temporal data and investigates its potential to improve weight prediction. The paper propose a model-agnostic time series forecasting framework for weight prediction with food intake. Our framework demonstrates significant improvements over existing methods in weight prediction performance.
2. Technical Correctness and Adequate Evaluation: The authors rigorously validate their framework by comparing it against established time series forecasting models, NLinear and iTransformer. This empirical assessment provides solid evidence of its improved performance, demonstrating technical soundness and practical relevance.
3. Clarity and Applications: The manuscript is well-structured and clearly articulates the problem, methodology, and findings. The DietDiary dataset creation adds significant value, facilitating further research in this domain.

**Limitations:**

1. Lack of Depth in Certain Aspects: The motivation of the paper is not clear, and the value and purpose of navigates weight prediction with dietary diary is not reflected.
2. As shown in Figure 11, although the predicted weight of the proposed method is better than NLinear, there is still a large gap between the predicted weight and the actual weight change. Do the tasks and methods presented in the paper make sense?
3. Why does the ablation experiment not include relevant experimental studies on the effectiveness of Unified Meal Representation Learning (UMRL)?

**Suitability:**

2

---

### Official Review · Reviewer_u3C3 · 2024-05-28

**Rating:** 4
**Confidence:** 3

**Summary:**

The paper "Navigating Weight Prediction with Diet Diary" embarks on a novel exploration into the predictive relationship between dietary habits and weight changes over time. It introduces the DietDiary dataset, which uniquely captures daily dietary logs and weight measurements, presenting a fresh avenue for empirical analysis in this domain. The central contribution is the innovative task of predicting future weight based on historical dietary data and the development of a model-agnostic time series forecasting framework. This framework is distinguished by its Unified Meal Representation Learning (UMRL) module, which adeptly extracts meal representations from diverse dietary inputs, and a diet-aware loss function that meticulously links dietary intake with weight fluctuations. The efficacy of this approach is validated through rigorous experiments, demonstrating its superiority over existing models by utilizing two advanced forecasting models, the NLinear and iTransformer.

**Strengths:**

The findings underscore the critical influence of integrating meal-specific data into weight prediction models, highlighting the improvement in predictive accuracy when dietary information is considered. The introduction of a multi-modal fusion strategy further refines the prediction capability, suggesting that combining different types of dietary data (images and textual ingredient labels) can significantly enhance the model's performance. The paper's methodical approach, from dataset creation through model innovation to detailed experimental validation, sets a high standard for future research in the intersection of diet tracking and weight prediction. This work not only advances the field of dietary data analysis but also opens up new possibilities for personalized health management and intervention strategies based on predictive modeling of weight changes.

**Limitations:**

The data looks sensitive. The author may declare how user privacy has been taken care of.

The data diversity should consider to be increased. It looks the image semantics are very similar related to time, which may cause a easy task for ML to solve.

More baseline models could be given and let us see how this dataset performs.

**Suitability:**

3

---

### Official Review · Reviewer_ruK9 · 2024-06-02

**Rating:** 5
**Confidence:** 3

**Summary:**

This paper addresses the gap in research exploring the impact of food intake on physiological indicators, such as weight, over time. It introduces the DietDiary dataset, which contains daily dietary diaries and corresponding weight measurements of 611 participants. The primary contribution of this research is the proposal of a novel task for weight prediction using dietary diaries. The task leverages historical food intake and weight data to predict future weights through a model-agnostic time series forecasting framework.

The proposed framework features a Unified Meal Representation Learning (UMRL) module to extract representations for each meal and a diet-aware loss function to associate food intake with weight variations. The framework is designed to be compatible with existing time series forecasting models. The performance of the framework is evaluated using two state-of-the-art time series forecasting models, NLinear and iTransformer, demonstrating superior performance compared to the original models.

**Strengths:**

This paper introduced a new task into food-related multimedia computing. Of course, human weight change does not depend on only dietary. It must depend on other enegry-comsumption activities. However, as the first step of this new task, this work is interesting, and the proposed dataset must be helpful for the community.

Since UMRL employs CLIP, both images and ingredient texts can be accepted as inputs. It is an interesting idea.

Basically the paper is written well and easy to read.

**Limitations:**

It is hard to evaluate if the estimate results are meaningfull or not. The statics of the dataset such as the average weight change over 3,5,7 days should be written. In this sense, adding MAPE (Mean Absolute Percentage Error) is helpful for the readers.

**Suitability:**

3

---

### Meta-Review · Area_Chair_2CMG · 2024-07-01

**Recommendation:** Accept (Oral)
**Confidence:** 4

**Metareview:**

All reviewers vote to accept the paper. The AC agrees with the reviewers and recommend acceptance.